# Noncoding RNA: An Insight into Chloroplast and Mitochondrial Gene Expressions

**DOI:** 10.3390/life11010049

**Published:** 2021-01-13

**Authors:** Asha Anand, Gopal Pandi

**Affiliations:** Department of Plant Biotechnology, School of Biotechnology, Madurai Kamaraj University, Madurai 625021, India

**Keywords:** ncRNAs, sRNAs, lncRNAs, chloroplast, mitochondria, gene regulation

## Abstract

Regulation of gene expression in any biological system is a complex process with many checkpoints at the transcriptional, post-transcriptional and translational levels. The control mechanism is mediated by various protein factors, secondary metabolites and a newly included regulatory member, i.e., noncoding RNAs (ncRNAs). It is known that ncRNAs modulate the mRNA or protein profiles of the cell depending on the degree of complementary and context of the microenvironment. In plants, ncRNAs are essential for growth and development in normal conditions by controlling various gene expressions and have emerged as a key player to guard plants during adverse conditions. In order to have smooth functioning of the plants under any environmental pressure, two very important DNA-harboring semi-autonomous organelles, namely, chloroplasts and mitochondria, are considered as main players. These organelles conduct the most crucial metabolic pathways that are required to maintain cell homeostasis. Thus, it is imperative to explore and envisage the molecular machineries responsible for gene regulation within the organelles and their coordination with nuclear transcripts. Therefore, the present review mainly focuses on ncRNAs origination and their gene regulation in chloroplasts and plant mitochondria.

## 1. Introduction

Technology improvements in the high-throughput sequencing of DNA and RNA and subsequent numbers of studies have provided extensive information on ncRNAs and their complex regulation networks in biological systems. In plants, more than 80–90% of transcripts are non-protein-coding RNAs [1,2], where a high number of these have been reported as potential regulatory molecules for various biological processes. With the availability of large data on plant ncRNAs, functional characterization reports revealed their pivotal potentials in controlling gene expression. However, investigations on plant organelles such as chloroplasts or mitochondrial genome-encoded ncRNAs are still sparse.

Generally based upon length, ncRNAs are categorized into three groups: small ncRNAs (sRNAs) with 18–30 nucleotides (nt) length, medium-sized ncRNAs (31–200 nt) and long ncRNAs (lncRNAs) (>200 nt). Further, based upon their origins, processing and mode of action, small ncRNAs are mainly divided into two classes, such as small interfering RNAs (siRNAs) and microRNAs (miRNAs, 18–21nt) [3]. siRNAs have been further sub-divided into two subclasses: transposable element-derived siRNA (24 nt) and phased siRNAs (phasiRNA, 21–23 nt or 24 nt). These two classes of small RNA (sRNA) have been widely reported from various plant species. Intriguingly, the lncRNAs which were initially recognized as transcriptional noises also contribute to gene regulation in plants by several pathways. Genes coding for sRNAs distribution on chromosomes arelinked with gene density [4,5]. In the distal euchromatic region, sRNA sequences are abundant, whereas they are abysmal in centromeric and pericentromeric regions. However, it has been shown that sequences complementary to sRNAs are also rich in centromeric and pericentromeric regions in some plants such as *Arabidopsis* [6], soybean [7] and cucumber [8]. The functions and biogenesis of sRNAs and lncRNAs have been studied in different plant species and reviewed extensively by various research groups [5,9,10,11,12,13,14].

Although recent advancements in genomic and transcriptomic analyses have enabled researchers to trace the presence of chloroplast-originated ncRNAs [15], only for few of them have functional studies been carried out [16,17]. Surprisingly, an abysmal number of ncRNAs has been registered from plant mitochondria, which is the opposite of that in animal mitochondria. RNA processing in both the semi-autonomous organelles is an important deciding factor for organellar gene regulation and protein profile [18,19]. It has been understood that the organelle genome undergoes complete transcription that leads to generation of an array of precursor RNA molecules that includes coding and noncoding transcripts [20,21]. Though most of the noncoding transcripts (intergenic, intronic or antisense) in organelles are considered as intermediate products of RNA processing, owing to less research on functional characterization, ncRNAs’ importance as a gene regulatory component cannot be denied. The organelles’ RNA metabolism machinery features are reminiscent of prokaryote ancestors due to the origination from them. Though organelles may have evolved from prokaryotes, the RNA metabolism and processing are complex and more developed. The reports on ncRNA-mediated regulatory events in the prokaryotic progenitors of chloroplasts and mitochondria or other eubacterial systems indicate speculation on the existence of ncRNAs in cell organelles [22,23]. Indeed, in animal mitochondria, it has been observed that ncRNAs mediate gene regulations inside sub-cellular compartments [24]. The intriguing studies on ncRNAs in mitochondria in animal systems raised questions concerning the exploration of plant ncRNAs originating from chloroplasts or mitochondrial genome- or nuclear-encoded ncRNAs imported to these cellular compartments. In this review article, we attempt to provide a comprehensive report on ncRNAs generated from chloroplasts and mitochondria of plant systems.

## 2. Chloroplast: Genome Structure and Regulation

Chloroplasts are a kind of a membrane-bound cell organelle, embedded within the outer and inner membranes. They range in size from 5 to 10 µm with highly variable shapes, existing as cup-shaped, spiral-shaped or star-shaped forms in algae, whereas they are saucer-shaped in bryophytes and higher plants [25]. The outer membrane is permeable for tiny organic molecules, whereas the inner membrane has several transport proteins which regulate the import and export of plastid-related proteins. The inner space of the organelle possesses an aqueous matrix called the “stroma” and a third membranous structure of “thylakoids” which acts as a center of photosynthesis. The stroma consists of ribosomes, metabolic enzymes, starch granules and multiple copies of the chloroplast genome attached to thylakoids [26]. From previous studies on chloroplast evolution, it was hypothesized that the green organelle tentatively originated from a cyanobacterial symbiont [27] which shares similar characteristics of the translation machinery to that in chloroplasts.

It has been estimated that one cell can contain ~10,000 copies of plastid DNA, attached to the thylakoid membrane along with the proteins and RNAs which arenamed nucleoids [28,29]. A plastid nucleoid consists of typically 10–12 copies of plastid DNA [29]. The chloroplast genome (plastome) is a circular doubled-stranded DNA, ranging between 120 and 160 kb. It may possess ~120–130 genes, which comprise 24 genes for rRNA and ribosomal proteins and nearly 30 genes for tRNAs and photosynthetic proteins. Six genes are essential for the ATPase subunits and four genes encode chloroplast RNA polymerase subunits. However, a large part of chloroplast proteins are derived from the nucleus [30,31]. Intriguingly, many genes of chloroplasts consist of introns that may cause generation of regulatory ncRNAs [32]. The plastid DNA actively participates in replication, DNA repair, transcription of genes of the photosynthetic machinery and gene expression; it is also involved in post-transcriptional regulation of gene expression in coordination with nuclear-encoded protein components [33,34]. Chloroplast gene transcription is conducted by two types of RNA polymerases: plastid-encoded RNA polymerase (PEP) and nuclear-encoded RNA polymerase (NEP), where most of the plastid genes have promoters for both types of polymerases [34]. RNA-binding proteins (RBPs), which are one of the important factors in organellar gene expression, are encoded by nuclear genes. These proteins carry out the maturation of the 5′ and 3′ ends of plastid transcripts and also participate in their decomposition. Generally, plastid RNA maturation involves conversion of specific cytidine residues to uridine residues that leads to restoration of a conserved amino acid. Pentatrico peptide repeat (PPR) proteins bind to a specific site on RNA to protect it against exonucleases [35,36,37]. There is an array of PPR proteins reported from plants that play roles in transcript processing. For example, CLB19 and CRR4 are important for the editing of *rpoA* (alpha subunit of plastid-encoded RNA polymerase), *clpP* (ATP-dependent caseinolytic protease) and *ndhD* (subunit of NA(P)DH dehydrogenase (NDH) complex) [35]. Tan et al. [38] explored a novel “*WSL*” (white stripe leaf) which codes for one chloroplast-localized PPR that is imperative for splicing of a chloroplast transcript, *rpl* 2 (ribosomal protein subunit L2). To substantiate *WSL* essentiality in plants, the mutation in *WSL* leads to sensitivity towards several stress responses. Similarly, apart from PPR, many other RBPs such as RLSB, an S-1 domain RNA-binding protein, and chloroplast ribonucleoprotein 31A are involved in regulation of organelle gene expression [39,40].

Plant development and stress tolerance capability are vastly determined by the photosynthetic yield occurring in the green organelle. Therefore, it is necessary to have a proper orchestral coordination between the nucleus and chloroplasts for unperturbed functioning of the organellar metabolic processes and, hence, the plant system. Besides carrying out photosynthesis, chloroplasts are crucial to conduct many other metabolic processes such as nitrogen and sulfur assimilation and nucleotides, amino acids and fatty acids synthesis. In addition, the green organelle also generates different kinds of secondary metabolites which act as signaling effectors during growth and development or in any stress condition [41]. Most of the plastid metabolic pathways are fully functional only in the presence of their nuclei-encoded protein components which are transported from the cytoplasm to chloroplasts in a regulated manner. Organelle coordination occurs through a stringent communication between chloroplasts and the nucleus and vice versa with the help of various signaling molecules.

Chloroplasts regulate the expression of nuclear genes via various plastid-originated signaling molecules, called “retrograde signaling”. Such signaling communication mostly takes place in response to developmental events or other external factors such as abiotic and biotic stress conditions that directly have an effect on photosynthesis. Chloroplast-to-nucleus signaling was first observed in barley mutant lines by measuring the reduction in nuclear genome-encoded plastid enzymes: phosphoribulokinase and D-glyceraldehyde-3-phosphate: NADP^+^ oxidoreductase (phosphorylating) [42]. These results led to speculate that less accumulation of the aforementioned plastid enzymes in the cytoplasm could be regulated through plastid-synthesized RNA. It was demonstrated that flowering transition under intense light can be regulated by chloroplast retrograde signaling. A chloroplast membrane-located transcription factor “PTM” (plant homeodomain (PHD)-type transcription factor with transmembrane domains) triggers transcriptional silencing of a nucleus-encoded floral repressor, “FLOWERING LOCUS C”, to regulate the timing of flowering transition [43]. Later, many more instances of retrograde signaling between organelles were reported [34]. Several chloroplast peptide molecules and many other plastid-generated metabolites are involved in different complex signaling networks inside the chloroplast or between the organelles. Based upon the environmental conditions, these signaling cascades control chloroplast homeostasis mainly by regulating chloroplast-related gene expression directly or indirectly which finally affects the plant growth and development [44,45]. On the other side, the nucleus-to-chloroplast communication process is named “anterograde signaling”, which has been studied extensively [46,47]. Recently, many drought-responsive lncRNAs in *Zanthoxylum bungeanum* (a shrub from the *Rutaceae* family) were found to be related to chlorophyll accumulation, chloroplast stability and photosynthesis. These nuclear-encoded lncRNAs were observed to be precursors for many miRNAs that control the expression of several chloroplast-associated targets such as *PHOT2*(Phototropin-2), *ARC3*(Protein accumulation and replication of chloroplast 3) and *UGP3*(UTP glucose-1-phosphate uridylyltransferase 3), among which *ARC3* and *UGP3* code for chloroplast-localized proteins [48]. In addition to protein factors and other secondary metabolites, ncRNAs have also been observed to show an impact on chloroplast gene expression or functions.

## 3. Noncoding RNAs in Chloroplasts as a Prospective Regulatory Molecule

Pervasive genome transcription occurring in chloroplasts causes random origins of transcription from genic and intergenic regions that leads to generation of unexpected transcriptional outputs [49]. Eventually, the subsequent RNA processing events become crucial in controlling plastid gene expression (reviewed by Del Campo [35], Zoschke and Bock [50], Robles and Quesada [51]). Interestingly, because of the complex maturation of RNA transcripts, aberrant termination might result in generation of several ncRNAs [52,53]. The ever expanding research on chloroplasts of different biological systems signifies the essentiality of ncRNAs as a possible new candidate for regulatory pathways [54,55,56] (Figure 1). Moreover, the presence of a large number of ncRNAs in bacterial systems allows exploring the plausible organellar ncRNAs and their roles in eukaryotic systems. Many ncRNAs are expressed in a tissue-specific manner or in a particular growth and development condition, or expressed at low level, which makes it a tedious job to capture them. Nonetheless, so far, a good number of reports over organellar ncRNAs from the plant kingdom have been published (Table 1).

A very first comprehensive report on ncRNAs from chloroplasts was provided by Lung et al. [68] by examining tobacco plants grown in light and dark conditions. Deep sequencing data analysis revealed that the size of sRNAs encoded by chloroplasts ranged between 19 and 53 nt. A set of novel ncRNAs (2.7%) was located in intergenic regions. Further, two other ncRNAs were mapped to the intronic region and one was antisense to the 3′ UTR of *atpE*. Afterwards, a study from sRNAs profiling under the tomato fruit ripening phase revealed a highly expressed cluster of sRNAs (named cluster “F”) in the chloroplast genome. However, the origin of sRNAs from “F” loci is yet to be deciphered due to the unannotated tomato genome which may have a part of the plastid genome via integration [69]. Out of 24, seven sRNAs loci overlapped with photosynthesis-related genes. Noticeably, a large part of this cluster, a 56 kb-long region containing sRNAs (derived from Photosystem I P700 chlorophyll a, chloroplast-derived DNA-dependent RNA polymerase subunits and ycf2) has interspersed inverted repeats which is a unique feature of the plastome [69]. Similarly, chloroplast sRNAs have been reported from many other crop plants such as melon, cabbage and barley [70,71,72]. sRNA sequencing data analysis in *Cucumis melo* showed that around 30,000 candidates mapped with the chloroplast genome, among which two clusters were found to be located within rRNA genes both in sense and antisense orientations [70]. One of the most abundantly present melon plastid sRNAs(a33_398374) showed sequence similarity with 900 plastid genome sequences from other plant species. The abundance of melon plastid sRNAs was observed to be tissue-specific, which exhibits their importance during photosynthesis. Furthermore, target prediction revealed methyltransferase as one of the targets for this sRNA; however, so far, no significant role of chloroplast sRNAs in transcriptional gene silencing has been reported.

Interestingly, yet another report from high and moderate temperature-treated Chinese cabbage exhibited the presence of plenty of chloroplast-generated sRNAs with 9–36 nt length [71]. The validation by Northern hybridization unambiguously confirmed that sRNAs originated from chloroplasts and not from the nucleus. Surprisingly, one of the small RNAs, “cs-RNA ig-1” (18 nt), matched with chloroplast ncRNA “ntc-1” from *Nicotiana benthamiana* [68], which reflects the conserved nature of organellar ncRNAs. Zhelyazkova et al. [72] predicted many ncRNAs from the barley plastid by analyzing the barley primary transcriptome. They observed many transcription start sites (TSS) located towards the upstream and downstream of annotated plastid genes and within intergenic regions as well. Further, Hachenberg and colleagues [73] compared the expression profile of functional sRNAs between transgenic barley overexpressing a drought tolerance transcription factor and a non-transgenic control plant. They observed that half of the sRNAs come from chloroplasts in both groups and among these, a highly abundant tRNA-derived sRNA matched with the 5′ end of the tRNA-HIS (GTG) chloroplast gene. The tRNA-derived sRNA was observed to match with the antisense strand of chloroplast genome-encoded 50s rRNA, which further supports the possibility of ncRNA-mediated regulation of chloroplast-generated transcripts. Recently, many ncRNAs deriving from tRNA with potential roles especially during stress conditions have been explored widely in various biological systems [74,75,76]. In *Arabidopsis thaliana*, 19–26 nt-long plastid tRNA-derived ncRNAs (plastid tRNA fragments) were detected through sRNA deep sequencing. More than 50% of the plastid tRNA fragments (ptRFs) arise from the 5′ end of tRNA with higher accumulation in cold-stressed than control plants. Surprisingly, Northern blot analysis of nucleus, cytoplasmic and plastid RNA fractions showed that plastid tRNA-derived ncRNAs are located in the cytoplasm but not in chloroplasts [77], which is contrasting to the tRNA (tRNA GTG)-derived ncRNA in barley [73]. It may be speculated that tRNA-derived sRNAs are translocated for efficient processing and regulation of gene expression during cold stress, which needs to be investigated further. Markedly, plastid-generated ncRNAs were efficiently detectable in plant systems under certain external environmental conditions such as heat stress or during important developmental stages, which indicates the potential role of organellar ncRNA in regulation of the proteomic profile inside the plastid [69,71].

Apart from higher plant chloroplasts, ncRNAs have also been identified in lower divisions of the plant kingdom such as algae. In a green alga, *Chlamydomonas reinhardtii*, the functional link between a nucleus-encoded RNA maturation factor, MMB1 (a member of the PPR family), and small RNAs mapping to the 5′ UTR of psbB and psbH was examined. It was concluded that these sRNAs from algal chloroplasts could be important as a landmark for identification of *cis*-acting elements essential for mRNA accumulation [59]. Similarly, clusters of organellar sRNAs (cosRNA) were observed through RNA and sRNA sequencing in the same unicellular algae by Cavaiuolo and coworkers [78]. Strikingly, cosRNAs were mapped with the 5′ end of 52 proteins coding RNAs, specifically coming from the mature 5′ proximity of the most abundant transcript “psbA”. These cosRNAs were suggested to be the footprint of nuclear genome-encoded maturation factor (M factor). A few antisense RNAs (asRNAs) loci from chloroplast genes were also seen which were speculated to regulate sense strand availability at the precise level of translation [16]. Recently, plastid transcriptomic analysis from a dinoflagellate, *Amphidinium carteraei*, showed the existence of noncoding sequences in the *psbA* and *petB*/*atpA* regions [79]. RNA hybridization and circular RT-PCR proved that the antisense RNAs of varying lengths in the plastid transcriptome possess a terminus that is different from their complementary sense strand, suggesting that these asRNAs follow a different terminal processing event. Therefore, it was proposed that asRNAs might have specific promoters located in reverse orientation or might be generated from transcription initiation events that do not require specific sequence motifs. The information on ncRNAs from the *psbA* and *petB*/*atpA* regions from various reports based on the chloroplast transcriptome provides a preliminary platform to analyze their functional importance in photosynthetic process regulation [79,80].

In spite of having the above-mentioned evidence on chloroplasts rRNAs from several transcriptome analyses in plants, many studies have proposed most of these organellar ncRNAs as a footprint of PPR proteins, RNA-binding proteins protecting plastid mRNAs’ termini from exonucleases [65,81,82]. An elaborative study from transcriptome data and cDNA end amplification results for selected sRNAs localized within intergenic regions of *clpP*–*rps12*, *atpH*–*atpF*, *cemA*–*petA*, *rpl18*–*rps20*, *rps7*–*ndhB*, *rps15* and *psaI*–*ycf4* in *Arabidopsis thaliana* revealed that most of the sRNAs reside close to the 5′ terminus and few reside at the 3′ end of mRNAs. It should be highlighted that these sRNA were highly conserved among the monocot and dicot plant chloroplast genome [81]. On the other hand, PPR family genes are also conserved among land plants, which show their functional importance in transcript protection [37]. An RNase protection test in the HCF152 (a PPR protein) mutant showed less accumulation of corresponding sRNAs, explaining their degradation by exonucleases. Similarly, a CCR2 (Chlororespiratory Reduction2) mutant plant (*Arabidopsis thaliana*) displayed absence of its respective 11 sRNAs [81]. In addition to this, Castandet et al. [83] used Terminome-seq, which revealed that most of the termini in *Arabidopsis* were located in intergenic regions and few were from coding sequences and intron regions. Interestingly, a large number of termini belonged to an sRNA that was a footprint of PPR proteins. Furthermore, the sequencing analysis showed the importance of the 3′ processing enzyme chloroplast polynucleotide phosphorylase (cpPNPase) in plastid RNA processing as they registered an increased number of altered termini in a *pnp 1-1* mutant plant [83]. Similarly, *Arabidopsis thaliana* chloroplast transcriptome analysis carried out previously by Hotto and coworkers [80] revealed the importance of cpPNPase in ncRNA accumulation and processing.

The length of organellar ncRNAs has a broad range. The sRNAs reported from chloroplasts can be lengthier than nuclear genome-transcribed sRNAs [68] or, in some instances, similar in size [15]. Until 2010, sufficient evidences on the functional importance of chloroplast sRNAs was available; however, reports on lncRNA were lacking. Georg et al. [84] reported a chloroplast long antisense RNA, i.e., asRNA_ndhB, in *Arabidopsis thaliana*, *Nicotianatabacum* and poplar, which covered the RNA editing site and group II intron splice acceptor site. The length of this antisense RNA varied from 400 to 650 nt and showed varied expression under different developmental stages. It was anticipated that asRNA_ndhB might be involved in transcript maturation or RNA stability. Interestingly, ectopic expression of chloroplast antisense RNA “AS5” in *N. tabacum* showed many folds of accumulation of the related sense transcript tRNA^Arg^ than 5S rRNA. Additionally, a reduction in ribosomal association of 5S rRNA and rbcL mRNA was observed, suggesting that AS5 may be needed for stability. Therefore, it was explained that this asRNA might play an important regulatory role in translation by controlling the 5S rRNA level [16]. To substantiate the above observation, similar results were examined on AS5 in an RRN1 (a 3′-to-5′ exoribonuclease) mutant *Arabidopsis*. Thus, it can be hypothesized that AS5 might be involved in inhibition of 5S rRNA maturation through endonucleolytic cleavage of precursor 5S rRNA by an RNaseIII-like enzyme [85]. Soon afterwards, 136 lncRNAs from *S. Miltiorrhiza* and a number of chloroplasts-related lncRNAs from *V. venifera* were reported [86,87]. Given the fact that chloroplasts possess two types of RNA polymerases (PEP and NEP) and a complex RNA processing system, chloroplasts have to have an extra regulatory system to control their transcriptome, which makes it clear that the ncRNA existence in chloroplasts cannot be overlooked.

## 4. Plant Mitochondria: Genome Structure and Regulation

Much like chloroplasts, mitochondria have their own genome and this semi-autonomous organelle originated from the symbiosis of an alpha-proteobacterium with a proto-eukaryotic host [88,89]. Mitochondria could be present from ~200 to ~600 copies per plant cell and their size ranges between 0.2 and 1.5 µm. They have a double-membrane structure and are quite an essential organelle for any eukaryotic organism as they provide the majority of cellular energy through oxidative phosphorylation. Thus, their functional coordination with the nucleus is very crucial which happens through various signal transducers [24,90,91]. Plant mitochondrial DNA is much larger than any other eukaryotic mitochondrial genome and has a large number of repeated sequences. It ranges between 70 and 11,000 kb, but its copy number is lower than in animals. Plant mitochondrial DNA undergoes frequent homologous recombination that causes evolution in mitochondrial DNA and hence it is considered to be less conserved [92,93]. It includes linear, circular, branched DNA strands surrounded by nucleoproteins and the complete structure is called a nucleoid [94,95]. Mitochondrial DNA contains genes for rRNAs, tRNAs, ribosomal proteins and subunits of respiratory chain protein complexes. Several open reading frames from plant linear mtDNA code for DNA and RNA polymerase and many other functional components, but, surprisingly, a very low number of coding elements have been observed compared to animal mtDNA [96]. In many plant species, a large number of circular mapping chromosomes are present which are considered to be “empty”, i.e., these are not coding for any functional transcripts but are, intriguingly, replicated and transmitted to the next population [97,98,99]. Therefore, their maintenance in the genome offers the question, what could be the reason for their existence? As of now, we have well explored ncRNAs as another layer of gene regulation in various organisms; one can speculate on whether these mini circles play any roles in generating functional ncRNAs. Strikingly, intergenic sequences in chloroplast and mitochondrial DNA are transcribed due to multiple transcription initiation sites in the genome. These sequences are speculated to contribute to producing regulatory ncRNA [68,100,101,102].

Similar to chloroplast DNA, mitochondrial DNA transcription appears to be a complex process and post-transcriptional events are governing factors in organelle gene regulation that mostly depends on inter-organelle coordination. RNA metabolism is mostly controlled by nuclear-encoded RBPs [19,51,90,103]. Transcription is mainly conducted by a nuclear-encoded phage-type RNA polymerase (RpoT gene family) and few components of the respiratory machinery are transcribed by mitochondrial “RpoTmp”, which is also present in chloroplasts [104]. Transcription in plant mitochondria starts from many promoters, whereas only three promoters are present in animals [105]. Transcriptional and post-transcriptional events in mitochondria are controlled by many nuclear-encoded *trans*-elements: PPR, RBPs, DEAD-box RNA helicases and mitochondrial transcription termination factors (mTERF) [51,103,106,107,108]. The functional importance of mitochondrial ncRNAs has been extensively studied in animals (reviewed by: Dong et al. [91], Dietrich et al. [102]). In many instances, it has been explored that these ncRNAs, in the case of animal systems, are either transcribed by the mitochondrial genome or transported to mitochondria from the nucleus and play important roles in organelle metabolic processes in normal and pathogenic conditions as well [100,109,110]. However, in the case of plants, there are only counted numbers of reports stating the presence of ncRNAs in this organelle which are speculated to have potential functions.

## 5. Noncoding RNAs in Plant Mitochondria as a Prospective Regulatory Factor

Studies on plant mitochondrial ncRNAs as regulatory molecules are ata naïve stage. Previously, a review by Dietrich et al. [102] on ncRNAs from organelles listed few examples of noncoding transcripts from plant mitochondrial origin, where most of these have proper editing sites which exhibit their probable requirement in gene regulation. To augment such observation, the mitochondrial transcriptome analysis of *Silene noctiflora* revealed 97 editing sites outside of the annotated sequences, which evidenced that noncoding transcripts coming from outside of genic regions are being processed by editing factors [99]. Furthermore, many sRNAs (17–25 nt) reads were coming from coding sequences of annotated mitochondrial genes. However, these sRNAs were considered as degrading products of mRNAs and rRNAs as these were overlapping and scattered all over the annotated regions [99]. Analogous to chloroplast cosRNAs, 24 nt-read length cosRNAs in *Arabidopsis thaliana* mitochondria have been identified using sRNA sequencing data and the bioinformatics pipeline “sRNA miner” [111]. Researchers observed that 45% of these cosRNAs were mapped with the nuclear genome, which might be due to the insertion of a large chunk of mitochondrial DNA into the nuclear genome [112]. In addition, it was seen that most mitochondrial cosRNAs aligned with the intergenic region and, mostly, these were overlapping with the 3′ end of the transcripts rather than the 5′ ends. The low abundance of 5′ end-aligning cosRNAs in mitochondria was proposed to be due to absence of 5′ to 3′ exonuclease activity [111]. 

Another study on cytoplasmic male sterility (CMS)-related genes in *Silene vulgaris* explored an 18 nt in length noncoding region from the transcriptome of floral bud tissue from a female plant (F) and hermaphrodites (H), named “Female Specific Noncoding RNA” (FSNR) [113]. This was asserted to be the first ever ncRNA sequence associated with plant CMS. CMS genes are located on the mitochondrial genome, whereas their expression is controlled by nuclear-encoded factors. A number of studies on CMS-related regulations in agricultural plants have explored nuclear-encoded miRNAs acting upon cytoplasmic targets and subsequently affecting metabolic functions in sterile lines [114,115,116]. However, so far, there is only one report on analyzing mitochondrial ncRNA for having any direct role in regulation of CMS gene expression [113]. Moreover, any mitochondria-localized nuclear-generated ncRNA involved in controlling CMS-associated gene expression is still to be explored.

The gene regulation through nucleic acid base modifications and its impact on plant development and tissue differentiation have been documented in various studies [117]. Interestingly, Murik et al. [118] reported multiple m^6^A modifications in mitochondrial intergenic noncoding transcripts from *Arabidopsis thaliana* and *Brassica oleracea*. The common sequence motifs analysis revealed a transcript with m^6^A modifications, NNHR^m6^AYSNN (where “H” corresponds to A, C or U; “R” corresponds to A or G; “Y” corresponds to C or U; “S” corresponds to C or G; and “N” corresponds to A, U, C or G), which shows similarity with the m^6^A motif present on nuclear-encoded mRNAs [118]. Hence, ncRNAs distributed all over mitochondrial transcripts may have a prospective significant role in regulation, as more than half of the m^6^A methylation modification was observed to happen in this region [118]. Therefore, it may be hypothesized that m^6^A modifications in ncRNAs may eventually affect their cognate targets. However, in this perspective, there are no proper reports available on regulatory roles of plant mitochondrial ncRNAs on target genes. Circular RNAs (circRNA), which are a type of long ncRNA, have been revealed as new emerging regulatory molecules in biological systems [119,120,121,122]. Recently, exonic mitochondrial circRNAs were identified in barley and *N. benthamiana*, where these were found to be involved in the regulation of the mitochondrial genome [121,123].

It is well known from several elaborative studies that animal mitochondrial ncRNAs involved in retrograde or anterograde signaling can play important roles in various gene regulatory networks. Therefore, it can be presumed that ncRNAs associated with plant mitochondria may also have such roles in plant systems (Figure 1). Given the fact that mitochondria are the source for aerobic energy generation and participate in maintaining plant growth and reproduction, it is considered by plant biologists to focus on regulatory networks involving mitochondrial ncRNAs which are either generated by mitochondrial DNA or imported from the nucleus. However, studying the existence of such organellar noncoding transcripts in plant mitochondria could be a challenging task, due to insertions of nuclear and plastid genomic origins [124]. Interestingly, there is evidence for transportation of nuclear DNA-encoded tRNAs to plant mitochondria [125,126]. Similarly, if there are chances for nuclear DNA-generated ncRNAs to be present in mitochondrial compartments, it can be expected that these ncRNAs may follow a similar kind of mechanism to enter plant mitochondria, but the underlying mechanism of their import to mitochondria needs to be studied.

## 6. Most Potential Factors Accountable for Chloroplast and Plant Mitochondrial ncRNAs Biogenesis

Recent searches from RNA sequencing data have proved that organelle DNA transcription is pervasive [49,127], giving an apparent reason for generation of a large number of ncRNAs. Moreover, full transcription of these organellar genomes leads to inefficient transcription termination; thus, organelle gene regulation mostly relies on various posttranscriptional events which involve various protein factors-mediated RNA processing and regulation of RNA stability [19,20,128].

High activity of NEP in the plastid during germination was suggested to cause complete transcription that leads to an elevated amount of antisense RNAs production in *Arabidopsis*. It was postulated that these antisense RNAs might be crucial for suppressing NEP-generated mRNAs during subsequent developmental stages where PEP activity becomes significant [57]. Interestingly, in barley, numerous TSSs were observed to be present within operons, intergenic and annotated complementary strand genes that were speculated to cause transcriptional uncoupling, leading to ncRNA generation [72]. The accumulation or production of these ncRNAs could be dependent either on chloroplast maturation or on plant developmental stages. Similarly, in *Chlamydomonas* chloroplasts, though few genes are cotranscribed, each individual gene also has its own promoter which might give rise to a cosRNAs TSS [78], as noticed in plant mitochondria [99]. Recently, a mitochondrial ncRNA, “FSNR”, which was considered to be associated with CMS in *Silene vulgaris*, was speculated to have its own promoter as it was not overlapping with any other transcribed region [113]. Thus, ncRNAs with a TSS could be transcribed by RNA polymerase, where RNA polymerase interaction with the promoter might depend on cooperative actions of other effectors that might be depending on physiological conditions or homeostasis of the organelle.

It is well known that ribonucleases and other RBPs play important roles in organellar RNA quality control. RBPs can possibly be leading a parallel pathway for regulatory sRNAs generation or their degradation if they are not required by an organelle. A chloroplast RNase mutation study in *Arabidopsis* suggested that there can be several different routes for ncRNAs biogenesis and degradation. CpPNPase is among one of the RNases that contribute in organelle RNA metabolism [129]. Changes in plastid ncRNAs abundance were observed in cpPNPase-lacking plants which prove its pivotal role in ncRNA accumulation and also its participation in their 3′ end maturation [80]. Another exoribonuclease (3′ to 5′), RNase R and RNase J have been suggested as important factors for chloroplast antisense RNA accumulation [80,128]. The deficiency of RNase E was suggested to destabilize as-psbK in *Arabidopsis* [80]. Thus, the nucleolytic action by these RNases might be one reason for the generation of organellar ncRNAs. Based upon previous observations, a possible PPR and a ribonuclease-mediated pathway for asRNA and ncRNA generation in chloroplasts are represented in Figure 2.

PPR proteins in chloroplasts and mitochondria are known as RNA protectors. They protect 5′ or 3′ ends of RNAs from exonucleolytic degradation caused by RNases, which are speculated to produce PPR footprints [81,130]. On the contrary, so far, such kind of sRNA has not been reported yet from plant mitochondria, presumed to be a different kind of RNA protection mechanism directed by mitochondrial PPR. It is expected that these kinds of sRNAs in chloroplasts could be deciding factors in mRNA stabilization and may compete for binding with their cognate PPR proteins [111]. Recently, in a CRY2 (a photoreceptor called cryptochrome)-overexpressing tomato plant, it was revealed that CRY2 can control abundance of plastid coding and noncoding transcripts [62].

Interestingly, Gomez and Pallas [131] studied the sub-cellular compartmentalization and replication of a pathogenic ncRNA (eggplant latent viroid) from the cytoplasm–nucleus–chloroplast. Therefore, it can be hypothesized that there could be a passage through which chloroplasts can acquire nucleus-encoded ncRNA from the cytoplasm apart from producing on their own. However, a further, more extensive investigation is needed. In addition, unlike in animal mitochondria [132,133], there are no detailed reports on the Argonaut (AGO) protein or any other similar kinds of proteins or precursor microRNAs in plant chloroplasts or mitochondria which could provide more information on the mechanism of their biogenesis. Interestingly, a systemic analysis on microRNA targets and the RNA-induced silencing complex (RISC) component in animal mitochondria revealed the colocalization of AGO proteins, microRNAs and target mRNAs on the outer membrane of animal mitochondria [134]. Thus, it can be speculated that RISC assembly may process sRNA maturation or regulate the translation of proteins associated with mitochondria. With several extensive reports on ncRNAs in plant organelles and some of evidence on their functional importance, it becomes interesting to explore further their pathway of biogenesis or their regulators. In this aspect, if other relevant accessory molecules/factors involved directly or indirectly in organellar ncRNA-mediated gene regulation networks are explored, it will help to obtain a roadmap of ncRNAs generation and their regulatory action.

## 7. Methods and Challenges to Identify Noncoding RNAs from Plant Organelles

With the recent progress in genome and transcriptome analysis, today, we have a large collection of genomic data for eukaryotic and prokaryotic organisms from different habitats. NGS technology has played a significant role in creating a huge depository of transcriptomic data related to protein coding transcripts or regulatory noncoding transcripts.

Mapping of the TSS on the plastid genome could serve as one useful method to predict and assert the presence of ncRNAs in chloroplasts. A novel differential RNA sequencing approach for plastids, isolated from *PEP* mutants, exhibited numerous TSSs on antisense strands or in intergenic regions of the plastome [72]. Recently, an established protocol, Terminome-seq, for chloroplast RNA isolated from leaves of wild-type and chloroplast RNA processing enzyme “cpPNPase”-deficient *Arabidopsis thaliana* also found numerous transcription start sites in intergenic regions, pointing towards the existence of several ncRNA promoters [83]. However, due to the multiple steps involved in this technique, the terminals of sRNAs with length less than 67 nt were not traced. However, the terminome sequencing method provides elaborate knowledge about 5′ and 3′ termini present in organelle primary and processed transcripts that might help in finding UTRs, promoter sequences and existence of PPR footprints.

In combination with high-throughput sequencing of theorganelle genome and transcriptome, computational analyses based on certain criteria have also been considered as significant tools to recognize functional organellar ncRNAs. However, the accuracy of prediction of ncRNAs based upon any algorithm typically depends on the information parameters chosen during analysis. Wu et al. [135] recently analyzed ncRNAs datasets for four different organelle genomes (chloroplast, kinetoplast, mitochondrion and nuclear genomes) using the “ID-SVM” algorithm. Based on the analysis, they proposed that the topology of the secondary structure, trinucleotides from three reading frames and motif information from ncRNA primary sequences are important parameters in ncRNA prediction. In addition, a bioinformatic pipeline, ChloroSeq, for analyzing chloroplast RNAs from strand-specific RNA-seq datasets, was recently created which was helpful in understanding the chloroplast transcriptome [136]. Another cosRNA-detecting software named “sRNA-miner” was developed to identify organelle sRNA from publicly available RNA-seq datasets [111]. In this bioinformatics pipeline, number of reads and sharpness of 5′ and 3′ ends can be adjusted. The principal parameters of this tool are sRNA species with less than 50 nt and sharp termini, which allows identifying sRNA from the region with a background of randomly decayed RNAs. YAMAT-seq (Y-shaped Adaptor ligated Mature tRNA sequencing) that was originally developed to perform high-throughput sequencing of tRNA based upon post-transcriptional addition of 5′-CCA-3′ at its 3′ end [137] was successfully used to detect other ncRNA species from animal mitochondria, with the same post-transcriptionally added signature secondary structure, i.e., 5′-NCCA-3′ or 5′-CCA-3′ protruding at the 3′ end [138]. A point may be considered here that only those RNAs which have a secondary structure can be identified and there can be chances that many ncRNAs with this kind of post-transcriptional modification may not go for reverse transcription during cDNA preparation. Despite this, YAMAT can be a promising protocol to be used to detect plant organelle ncRNAs.

Another approach to analyze the functional importance of any transcript in mitochondria and chloroplasts could be to knockdown the organellar transcript by transporting a customized RNA inside the organelle. In various studies, it has been shown that tRNA can carry a passenger RNA molecule to mitochondria [139,140]. In this context, a tRNA mimic, PKTLS (mimic of tRNA^Val^ that was developed from the 3′ end of *Turnip yellow mosaic virus* (TYMV) genomic RNA), was demonstrated for serving the purpose of passenger RNA (ribozyme Rzatp9) translocation from the cytoplasm to mitochondria in tobacco cells and the *Arabidopsis thaliana* plant [141,142]. In conclusion, knocking down any important organellar mRNAs might give a proper landscape of a changed transcriptome profile within the organelle and change in inter-communication between the nucleus and mitochondria or chloroplasts. Expectedly, this kind of chimeric RNA tailor designed to target organelles can be an advanced tool in understanding gene regulation inside organelles, especially with respect to organellar ncRNAs. However, designing such chimeric ribozyme with high specificity to its related mRNA could be a critical task.

Today, we have various techniques to study ncRNA-mediated gene regulation in multicellular organisms. Nonetheless we need to conduct a series of experiments in various aspects to validate the genuineness and functional importance of organelle-generated ncRNAs, as it is known that organelle genome fragments are also present in the nuclear genome. In addition, organelle gene expression is partially regulated by the nucleus. Therefore, conducting a comparative study by taking purified organellar RNA/DNA and total cellular RNA/DNA under various physiological conditions, as has been conducted previously in many studies, can be an appropriate method to gain an unambiguous picture. However, a signal-based RNA detection assay or RNA imaging could be a rapid method to trace organellar ncRNAs. Many researchers have recently explored the scope of utilizing the highly sensitive CRISPR (clustered regularly interspaced short palindromic repeats)-based tools in RNA detection or sub-cellular localization of RNA molecules [143]. For an example, Wang and coworkers [144] demonstrated a method called LiveFISH (CRISPR Live Cell Fluorescence in situ Hybridization) for live cell imaging of genome editing and transcription simultaneously. They used different fluorophores labeled Cas9(CRISPR-associated endonucleases) guide RNA (gRNA)and Cas13d gRNA for detecting DNA and RNA, respectively. These CRISPR-based techniques could be helpful in detection of ncRNA in sub-cellular compartments if a highly sensitive CRISPR-Cas-based probe could be developed for short-length sRNAs, as there may be chances of having a high signal-to-background ratio during microscopic visualization. Various potential methods that can be used to study organellar ncRNAs are listed in Table 2.

## 8. Concluding Remarks

In this review, we have explained the current scenario of different noncoding RNA molecules with respect to their generation and functional potential in plant sub-cellular compartments. There are enormous reports available on nuclear genome-transcribed functional ncRNAs in different plant species stating their roles in regulation of growth and development and their involvement in regulation of stress-responsive factors. On the other side, limited research has been conducted on ncRNAs from two other plant cell organelles, i.e., chloroplasts and mitochondria. Over the past decade, a good number of transcriptomic studies from chloroplasts have shown the presence of enormous ncRNAs, though further validation is required. Several plastid-encoded asRNAs were proposed to play an important role either as a negative or positive regulator of sense strands. Additionally, several cosRNAs reported ranging from algae to higher plant chloroplasts and mitochondria are mapped with regulatory 5′ or 3′ UTRs of coding sequences, which shows their possible importance in post-transcriptional regulation of the related genes. In addition, ncRNAs accumulation seems to be dependent on different developmental stages of plants and environmental conditions.

Relaxed transcription of the organelle genome generates many intergenic and antisense transcripts along with coding transcripts, and depending upon organellar metabolic activities, the noncoding transcripts are degraded by different nuclear-encoded ribonucleases. However, many RNA sequencing studies in normal and ribonuclease mutant plants have proposed the significance of ncRNAs in regulation of organelle genes. So far, it is established from previous studies that RNA metabolism in chloroplasts and mitochondria is mostly regulated by RBPs and ribonucleases; on account of actions by these two kinds of proteins, RNA footprints are produced which might mostly be destined to be degraded. Strikingly, the RNA fragments generated due to transcript protection action by the PPR protein were proposed as potent competitors of cognate mRNAs for binding with PPR; in other words, these may be one of the deciding factors for mRNA translation [112]. Detection of ncRNAs has been always a tedious task due to their low expression level and this makes it more difficult to study them from sub-cellular organelles. However, there are many appreciable attempts at plant organellar ncRNAs investigation that have been carried out by various plant biologists, which could provide the base for more advanced research on ncRNAs [86,113,123]. In brief, tracing the ncRNA from two vital cellular organelles, i.e., chloroplasts and mitochondria, particularly from plants grown in abiotic or biotic stress conditions, might guide researchers towards an additive alternative to improve crop quality with respect to adaptation to adverse conditions and crop yield.

## Figures and Tables

**Figure 1 life-11-00049-f001:**
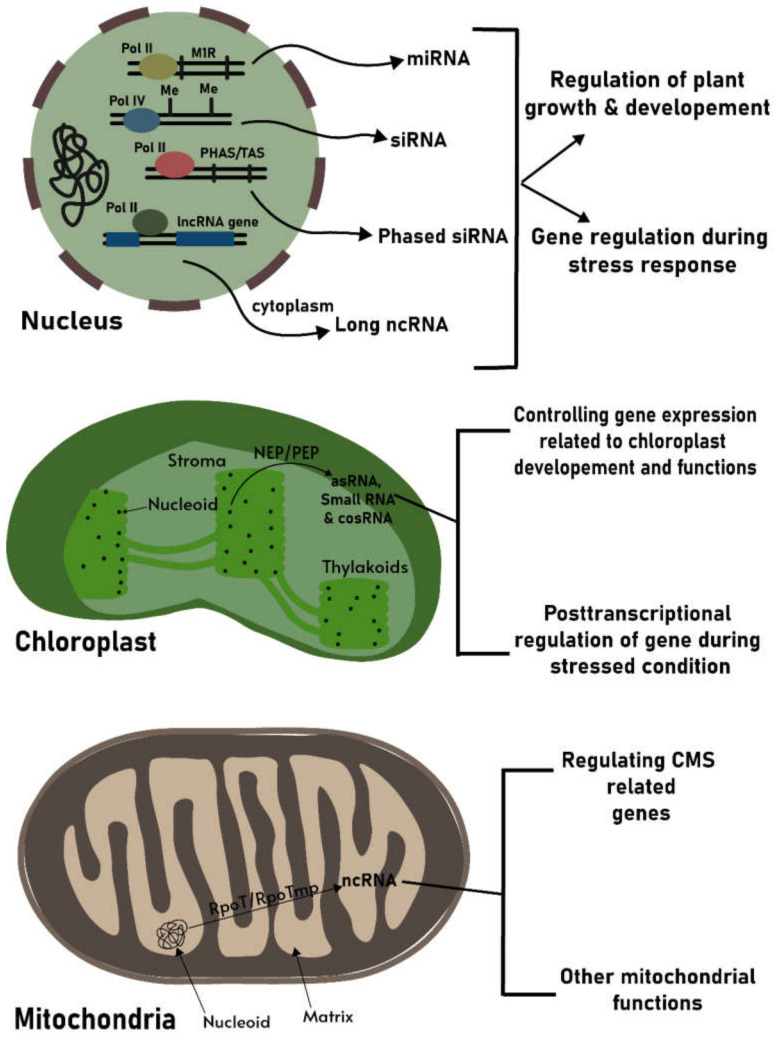
A summarized view of noncoding RNAs (ncRNAs) functions in plant organelles. In the nucleus, polymerase II- and IV-mediated transcription generates ncRNAs that are important regulators of plant growth and development in normal and stress conditions as well; in chloroplasts, PEP/NEP-generated ncRNAs might participate in post-transcriptional regulation of organelle genes and regulate chloroplast development; in mitochondria, ncRNAs are generated by RpoT/RpoTmp polymerases, where these can be involved in regulation of cytoplasmic male sterility (CMS)-related genes or other important mitochondrial functions. MicroRNA gene (MIR); methylated (Me); phased siRNA gene (PHAS); trans siRNA gene (TAS); long noncoding RNA gene (lncRNA gene); Plastid encoded RNA polymerase (PEP); Nuclear encoded RNA polymerase (NEP).

**Figure 2 life-11-00049-f002:**
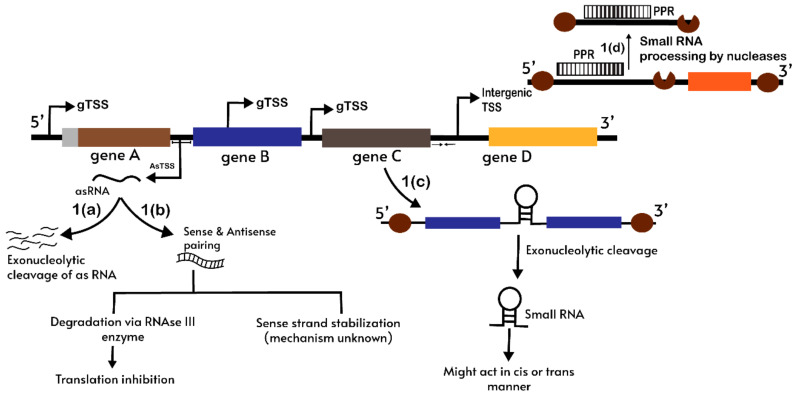
A schematic representation of the pathway for generation of ncRNA in chloroplasts. 1(a) asRNA are transcribed from asTSS and these asRNA are degraded by the action of exonucleolytic enzymes. 1(b) asRNA pairing with a sense strand which either causes translation inhibition or takes part in mRNA stabilization via an unknown mechanism. 1(c) Gene-specific transcription start site (TSS) may lead to sRNA generation through stable transcript structures formed at UTR regions where RNase movement is blocked. 1(d) ncRNA may also arise from overlapping 5′ and 3′ UTR regions of polycistrons that are protected from plastid RNases by PPR proteins. The brown round structures are 5′ to 3′ and 3′ to 5′ exonucleases; the brown colored half-round structure is an endonucleolytic enzyme. Transcription start site (TSS); Untranslated region (UTR).

**Table 1 life-11-00049-t001:** Noncoding RNAs from chloroplasts.

Noncoding RNA. (Name, Numbers)	Length (nt)	Source (Plants/Algae)	Corresponding Sequence	ncRNA Detection Methodology	Reference
Antisense RNA	-	*Arabidopsis thaliana*	Opposite strand of rbcL, psbA and rpl33	Macro array analysis	[57]
Small ncRNAs, 27	20–24	*Deschampsia antarctica* Desv.	Located near 5′/3′ ends of genes, i.e., psbH-petB, atpH 5′ end, ndhB 5′ end, and petD_rpoA	RNA sequencing data analysis	[58]
Short RNA	17–24	*Chlamydomonas reinhardtii*	5′ end of psbB and psbH	High-throughput RNA sequencing	[59]
rnpB	-	*P. purpureum*	-	Plastid genome sequencing	[60]
Antisense RNAs	-	*Karenia mikimotoi* (dinoflagellate)	psbA, psbD, ycf4, rps13 and rps11	cDNA libreary sequencing, RT-PCR, cloning of 5′ end obtained from RACE	[61]
miRNA, 79	22	*Solanum lycopersicum* (tomato)	Intergenic and intronic regions; Tpr and PPR, lucine rich repeat gene	miRBASE Search tool against tomato leaf trancriptome data	[62]
circRNAs	-	*Arabidopsis thaliana*	Exonic	Based on ncRNA sequencing dataset	[63]
ncRNA in IGS (intergenic spacer) region	-	*Solanum dulcamara*	Intergenic regions	RNA sequencing library from NCBI short read archive for *Solanum dulcamara* (SRR2056039)	[64]
sRNA	19	*Zea mays* ssp. mays	3′ UTR of *psbH* gene	sRNA sequencing, Northern blot	[65]
circRNAs	-	*Arabidopsis thaliana*	Intronic, exonic, intergenic	circRNA sequencing	[66]
ncRNA	-	*Ficus tikoua*	-	CP genome sequencing and assembly	[67]

**Table 2 life-11-00049-t002:** List of potential methods to detect organellar ncRNA.

Potential Methods to Detect Organelle ncRNA	Description	Reference
Differential RNA sequencing (dRNA-seq)	dRNA-seq was used to discriminate primary transcript from processed 5′ ends by sequencing differential cDNA library	[72]
Terminome-seq	Library synthesis was conducted by using the Illumina TruSeq Small RNA library preparation kit, where the RNA populations containing a 5ʹ phosphate and 3ʹ hydroxyl group were captured	[83]
Algorithm-based ncRNA prediction	Secondary structure; three reading frames with triplates of nucleotide and motif information from ncRNA primary sequences were used as parameters to recognize ncRNAs via ID-SVM algorithm	[135]
ChloroSeq	A bioinformatics approach that is used to study chloroplast RNAs from strand-specific RNA-seq datasets	[136]
sRNA-miner	Publically available RNA-seq datasets were used to detect organelle ncRNAs	[111]
YAMAT-seq	ncRNA species with post-transcriptionally added signature secondary structure, i.e., 5′-NCCA-3′ or 5′-CCA-3′ protruding at 3′end can be detected	[137]
“PKTLS” (a chimeric transfer RNA-like structure with catalytic activity)	Knockdown of important organellar transcript	[141]
Fluorophore-labeled CRISPR-Cas-based probes	RNA imaging in live cells	[144]

## Data Availability

Not applicable.

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
