# Peer review of "Noncoding RNA: An Insight into Chloroplast and Mitochondrial Gene Expressions"

_life, 2021, doi:10.3390/life11010049_

Round 1
Reviewer 1 Report
Comments for authors
This review article aims at summarizing the researches that have been done so far on the topic of non-coding RNAs (ncRNAs) originating from the genome of plant semi-autonomous organelles i.e., chloroplasts and mitochondria. As is mentioned in the article, while there are accumulating reports available on ncRNA in nuclear genome, less attention has been paid to ncRNAs from those other plant cell organelles, therefore, this article would provide a unique insight into the function of organelle ncRNAs in inter-communication between nucleus and mitochondria or chloroplast especially in terms of gene regulation. The manuscript is well documented with 8 sections, including an overview of genome structure and regulation in chloroplast and mitochondria, several examples of organelle ncRNAs in some plant species, and possible approaches to identify ncRNAs in plant organelles by focusing on ncRNAs origination and maintenance in chloroplasts and mitochondria. I think there is basically no problem to publish the article in Life in the current form, but I have some suggestions as described below, especially regarding functional aspect of those organelle ncRNAs. I request the authors to add some revisions by referring the following comments. I hope this revision will be of help to improve this article.
- Generally speaking, information of length distribution, conservation, changes in expression in the context of development, stress response etc., compared with those of mRNAs are often described in review papers. I would prefer to add such information to this article as well if there are any reports analyzing those kinds of statistics. For example, do ncRNAs from the chloroplast genome tend to be shorter than those from nuclear genome? How about their conservation when comparing with other species’ organelle genome? Any reports on describing changes in the expression of organelle ncRNAs in response to photosynthesis, nutrient stress, energy limitation, etc.? Ref 68 looks like one of those reports in which RNAs from tobacco chloroplast were analyzed in dark and light conditions.
- Is there any reason for not including a list of mitochondrial ncRNAs, while there is a list of chloroplast ncRNAs as Table 1?
- I assume it is relatively easy to find miRNAs from the genome sequence by using computational search tool, however, only a few information is found in the article about miRNAs originating from organelles. According to Table 1, there seems to be at least one report which attempted to search chloroplast miRNAs (ref 62). If it doesn’t mean miRNAs in those two organelle genome are pretty rare, it would be better to add more information in terms of identification and function (or predicting target genes) of organelle miRNAs. Since miRNA is the best studied ncRNA species so far both in animal and plants, it looks a bit wired if there is less explanation for this.
- As for figure 2, I’m curious about how the small RNAs produced in the organelle can function. More specifically, as far as I understand, RISC formation is required to activate those small RNAs, therefore RISC forming proteins such as AGOs are required to exist in the organelle or the small RNA should be exported to the cytosol, though I am not sure if the organelle genome encodes RISCs. If there is a published model that can explain this point, I do recommend to refer the model either in the manuscript or in the figure, as well as reflect the model in figure 2.
- In the section 7, Methods and challenges to identify noncoding RNAs from plant organelles, the authors introduced a way to analyze function of organelle ncRNAs by using PKTILS mediated translocation of target RNA from cytosol to the organelle. Are there any researches that used this approach to analyze organelle ncRNAs either in animals or plants? If yes, it is better to add those references with a brief description. Related to functional analysis of organelle ncRNAs, the authors pointed out its difficulties because “organelle genome fragments are also present in nuclear genome (line 502)”, I would propose imaging-based approaches may also be help to overcome the problem, such as CRISPR mediated imaging of transcription, though there might remain several things we have to improve in order to make it possible to get high-resolution images of transcription in plant organelles by microscopic analysis.
- Although this article mainly focused on origination and regulation of organelle ncRNAs, the authors could provide a bit more information of function of each type of ncRNA described in the article. Lines 389-403 at least partially covers this point, but I would require to add some more examples to enable readers to imagine the importance of organelle ncRNAs to organize the regulation of gene expression both in the organelle and the nucleus. For example, how the m6A-modified ncRNAs transcribed from intergenic regions of organelle genome could affect the gene regulation (line 359)? Does “45% of cosRNAs mapped to nuclear genome (line 340)” imply possible effect of those ncRNAs on the expression of nuclear genome encoding genes? Does it imply tRNA derived sRNAs could also affect the translation in the chloroplast (line 211)? If there is a separated section describing proposed models of the molecular function of each organelle derived ncRNA, it would make it easy to access to such information.
In addition, the followings are minor things to be corrected:
L39: Need a space before “These”. Besides this, please make sure if use of space and font-type such as italic or capitalization is correct in all the manuscript.
L71: “μ” -> “μm”
L361: NNHRm6AYSNN needs a description of what each symbol stands for, e.g. ‘R’ corresponds to A or G.
Figure 1:
The arrowheads pointing the words nucleoid or matrix is better to be opposite. Otherwise it looks like, for example, the nucleoid is exported to cytosol. All the abbreviations in the figure have to be described in the caption: MIR, Me, ITGS, PHAS, TAS, and so on. I’m wondering why the types of ncRNAs are specified in case of nucleus, partially specified by annotating as “asRNA, small RNA, and cosRNA” in case of chloroplast, and not specified and just annotated as “ncRNA” in case of mitochondria. Do the authors want to mention the possible difference in the type of ncRNA species in each organelle? Another thing I would point it out is text in the mitochondria is hard to see.
Author Response
Dear Editor,
We are enclosing herewith the revised manuscript entitled “Noncoding RNA: An insight into chloroplast and mitochondrial gene expressions” by Asha Anand and Gopal Pandi. We have gladly received the reviewer’s comments and made the changes suggested by the Editor and the Reviewers. We have incorporated all your suggestions into the development of our manuscript. As suggested, we have highlighted the changes in the manuscript by using track changes. We are thankful to the Editor and Reviewers for their valuable suggestions in upgrading the quality of the manuscript. We look forward to receiving your kind reply.
Thanks and regards
Dr Gopal Pandi,
Department of Plant Biotechnology
MKU, India.

Reviewer 2 Report
Please, see comments throughout the attached manuscript.

Author Response
We are enclosing herewith the revised manuscript entitled “Noncoding RNA: An insight into chloroplast and mitochondrial gene expressions” by Asha Anand and Gopal Pandi. We have gladly received the reviewer’s comments and made the changes suggested by the Editor and the Reviewers. We have incorporated all your suggestions into the development of our manuscript. As suggested, we have highlighted the changes in the manuscript by using track changes. We are thankful to the Editor and Reviewers for their valuable suggestions in upgrading the quality of the manuscript. We look forward to receiving your kind reply.
Thanks and regards,
Dr Gopal Pandi,
Department of Plant Biotechnology
MKU, India.

Round 2
Reviewer 1 Report
I think the latest version of manuscript is ready for publication. Although it was not due to my suggestion, I love the idea of adding table 2 which summarizes potential methods for organelle ncRNA detecting. I hope this review article will stimulate the field of plant ncRNA research.